# Shigella Outer Membrane Vesicles as Promising Targets for Vaccination

**DOI:** 10.3390/ijms23020994

**Published:** 2022-01-17

**Authors:** Muhammad Qasim, Marius Wrage, Björn Nüse, Jochen Mattner

**Affiliations:** 1Department of Microbiology, Kohat University of Science and Technology, Kohat 26000, Pakistan; qasim89@gmail.com; 2Mikrobiologisches Institut-Klinische Mikrobiologie, Immunologie und Hygiene, Universitätsklinikum Erlangen and Friedrich-Alexander Universität (FAU) Erlangen-Nürnberg, 91054 Erlangen, Germany; bjoern.nuese@uk-erlangen.de (M.W.); marius.wrage@uk-erlangen.de (B.N.)

**Keywords:** outer membrane vesicles (OMVs), *Shigella*, vaccination

## Abstract

The clinical symptoms of shigellosis, a gastrointestinal infection caused by *Shigella* spp. range from watery diarrhea to fulminant dysentery. Endemic infections, particularly among children in developing countries, represent the majority of clinical cases. The situation is aggravated due to the high mortality rate of shigellosis, the rapid dissemination of multi-resistant *Shigella* strains and the induction of only serotype-specific immunity. Thus, infection prevention due to vaccination, encompassing as many of the circulating serotypes as possible, has become a topic of interest. However, vaccines have turned out to be ineffective so far. Outer membrane vesicles (OMVs) are promising novel targets for vaccination. OMVs are constitutively secreted by Gram-negative bacteria including *Shigella* during growth. They are composed of soluble luminal portions and an insoluble membrane and can contain toxins, bioactive periplasmic and cytoplasmic (lipo-) proteins, (phospho-) lipids, nucleic acids and/or lipopolysaccharides. Thus, OMVs play an important role in bacterial cell–cell communication, growth, survival and pathogenesis. Furthermore, they modulate the secretion and transport of biomolecules, the stress response, antibiotic resistance and immune responses of the host. Thus, OMVs serve as novel secretion machinery. Here, we discuss the current literature and highlight the properties of OMVs as potent vaccine candidates because of their immunomodulatory, antigenic and adjuvant properties.

## 1. Introduction

*Shigella* species invade the gut-lining epithelium and cause shigellosis, also known as bacillary dysentery. Infections with *Shigella* are among the leading causes of bacterial diarrhea world-wide, [1] and the mortality rate of shigellosis can be high, particularly in epidemic infections with *Shigella* (*S.*) *dysenteriae* serotype 1 or endemic *S. flexneri* infections in children living in areas with a high prevalence of malnutrition [2]. Although shigellosis is a global health problem in all age groups, endemic infections in children, particularly in developing countries with poor sanitation and/or with poor personal hygiene, constitute the main disease burden [3].

*Shigellae* are transmitted via water, food or via the fecal–oral route. The infectious inoculum is low, which further facilitates the spread of this very contagious bacterium from one person to another. Even after diarrhea stops, an individual person can shed and transmit *Shigellae* for several weeks. Thus, low hygiene and sanitation standards facilitate the person-to-person spread of *Shigella* spp. Furthermore, therapeutic options become increasingly limited, as *Shigella* strains rapidly develop resistance to commonly used frontline antibiotics including beta-lactame antibiotics, fluoroquinolones, tetracyclines, and aminoglycosides, and even become multi-drug resistant (MDR) [4,5]. As there are multiple serotypes associated with illness, repeated infections in one individual are common. However, the decrease in the incidence of disease with increasing age suggests that protective immunity develops [2,6,7]. Thus, vaccination might be an adequate tool to restrain the person-to-person spread of *Shigellae* and/or the severity of shigellosis in any one individual.

## 2. *Shigella* spp. and *Shigellosis*

*Shigella* spp. are Gram-negative, non-motile, and facultative anaerobic rods belonging to the family of Enterobacteriaceae. Based on 16S rRNA sequence analyses [8,9], we know nowadays that *Shigella* spp. are closely related to *Escherichia* (*E.*) *coli* strains. Indeed, today’s commensal and pathogenic *E. coli* and *Shigella* variants share common progenitor strains [10]. Thus, *Shigellae* are considered as *E. coli* strains that have acquired a specific set of genes that contribute to their specific pathogenesis and clinical pathology [9,11]. However, historically, *S**higella* spp. and *E.* *coli* strains have been classified as separate species based on their biochemical characteristics, serological typing, and clinical relevance [12,13,14].

The genus *Shigella* itself is sub-divided into four species: *S. dysentriae*, *S. flexneri*, *S. boydii*, and *S. sonnei*. Each species is composed of different serotypes based on the structure of their lipopolysaccharide (LPS) antigen, the so-called O antigen, repeats [15]. At least 50 different serotypes exist [15,16]. The identification of their distribution is pivotal for the success of vaccination strategies as acquired antibody responses to *Shigellae* are serotype-specific [6].

Shigellosis can present with versatile symptoms ranging from acute watery diarrhea to fulminant dysentery, accompanied by fever and abdominal cramps. While *S. boydii* and *S. sonnei* usually cause a relatively mild illness, which includes watery or bloody diarrhea, *S. flexneri* and *S. dysenteriae* are predominantly responsible for endemic and epidemic shigellosis in developing countries, with high transmission rates and significant fatality case rates. Particularly, *S. dysenteriae* serotype 1 causes severe disease and might be associated with life-threatening complications [2].

The ability of *Shigella* spp. to cause this wide range of symptoms has been linked to different virulence factors, which are encoded on chromosomal pathogenicity islands and the virulence plasmid [17]. These virulence factors include the type III secretion system, lipopolysaccharide (LPS) or the Shiga-toxins among others and mediate inflammation and enterotoxic effects, subvert the host cell structure and function, and suppress protective immune responses [17].

The distribution of *Shigella* species varies among geographical regions. *S. dysenteriae* is a major causative agent of severe epidemic disease in less developed areas while *S. flexneri* and *S. sonnei* occur primarily in developing and developed countries [18,19].

Indeed, the persistent colonization of local areas by sub-lineages of *S. flexneri* may have contributed to the species’ long-term success [20]. *S. boydii* is confined to the Indian subcontinent, and *S. sonnei* occurs in both transitional and developed countries [17,18]. Stratified analyses indicated a decrease in the prevalence of *S. flexneri* cases and an increase in the prevalence of *S. sonnei* cases concurrent with the rapid economic growth experienced by China in recent years [21], as observed in many other emerging countries [22,23,24]. Thus, as acquired immune responses are serotype-specific [6], the local circulating *Shigella* strains and serotypes need to be identified in order to develop successful vaccine candidates.

## 3. Vaccination Approaches

Several approaches have been employed to obtain effective vaccine candidates against shigellosis. These included (sub-) cellular vaccines, formalin-inactivated whole *S. sonnei*, glycoconjugates, subunit candidates, live attenuated deletion mutants, conjugated detoxified polysaccharide parenteral vaccines or conjugated synthetic carbohydrates among others [6,25,26]. Unfortunately, despite the broad variety of strategies applied and the number of potential candidates identified, there is not yet a convincing or licenced vaccine against shigellosis available. Although some vaccines have been evaluated in clinical settings, further pre-clinical and clinical studies are urgently required once the safety, immunogenicity, and protection of promising vaccine candidates have been confirmed in challenge models. Owing to the wide range of *Shigella* serotypes and subtypes, there is a need for the development of a multivalent vaccine that represents the prevalent species and serotypes [27]. Due to their characteristic assembly of antigenic structures and virulence factors, the outer membrane vesicles (OMVs) of *Shigella* are promising targets for vaccination in this context. Furthermore, the possibility of mixing OMVs from multiple serotypes presumably allows the induction of a multi-serotype immunity. Last, OMV-based vaccines can be applied orally, and thus, might overcome the ineffectiveness of parenteral vaccines to stimulate a mucosal immune response.

## 4. Outer Membrane Vesicles (OMVs) as Targets for Vaccination

Outer membrane vesicles (OMVs) are approximately 50–300 nm large vesicles released by Gram-negative bacteria during growth in vivo as well as in liquid and solid culture in vitro [28,29]. They consist of many potential antigenic structures including proteins, lipoproteins, lipopolysaccharides (LPS), toxins, nucleic acids, lipids, and phospholipids derived from the periplasm and from the outer bacterial membrane [28,29] (Figure 1). Thus, they are promising targets for vaccination.

OMVs presumably contribute to the stress response of bacteria, the removal of misfolded proteins, inter-bacterial communication, the transmission of virulence factors, and the delivery of molecules and toxins [30]. Furthermore, they facilitate horizontal gene transfer, the acquisition of nutrients, and the formation of biofilms. In addition, they help in the establishment of a colonization niche, protection against antibiotics, and neutralize anti-microbial peptides [29,31]. Thus, OMVs improve bacterial survival on the one hand, while on the other, they also enhance immune responses in host cells, communicate with epithelial and immune cells, and modulate host-pathogen interactions, making them attractive targets for an immune attack (Figure 2).

*Shigella* OMVs contain virulence factors that facilitate the adhesion of *Shigella* to colonic epithelial cells, the inter- and intra-cellular spread, biofilm formation, and immune evasion. Thus, OMV-based vaccines not only ameliorate the severity of disease, but also inhibit the person-to-person spread.

Thus, OMVs represent promising tools for various biomedical applications including vaccination, immunotherapy, drug delivery and as vehicle for infection control. Indeed, due to their immunogenic nature, stability in various chemical and physical conditions, and low cellular toxicity, OMVs are excellent vaccine targets [32]. Thus, OMV-based vaccines against many bacteria including *E. coli*, *Salmonella*, *Vibrio* and *Campylobacter* spp., *Helicobacter pylori* as well as *Neisseria* and *Shigella* spp. have been intensively studied [33,34,35,36,37]. Although OMVs are considered as cost-effective, non-living vaccine candidates against *Shigella* [38], detailed knowledge about the biology of OMVs on immune-modulation and bacterial pathogenesis is missing. Thus, the objective of this review is to summarize the current knowledge on the structure and chemical composition of OMVs as well as on their biological role. Furthermore, we discuss the virulence factors of *Shigellae,* and in particular, of *S. flexneri* and their interaction with the immune system of the host. Last, we describe OMV production methods in order to obtain sufficient amounts of antigens.

## 5. Immunological Responses Triggered by *Shigella* OMVs in Infection Models

A number of studies have investigated the immunogenic properties of OMVs from *Shigella* species in animal models (Table 1).

In one study, an intranasal immunization with OMVs protected 100% of nine-week old, female BALB/c mice against an infection with 1 × 10^7^ CFUs *S. flexneri* 2a 35 days later [39]. The OMVs from *S. flexneri* 2a thereby induced serotype-specific protective IgG1 and IgG2a antibodies. In another study, about 50% of OMV-immunized female BALB/c mice survived a lethal dose of 4 × 10^6^ CFUs of *S. flexneri* 2a [40]. Importantly, however, OMV vaccination did not only induce sero-type specific protection; for example, the oral administration of an OMV mixture from *S. dysenteriae* 1, *S. flexneri* 2a, *S. flexneri* 3a, *S. flexneri* 6 and *S. sonnei* to 7-week female Swiss mice elicited the secretion of protective cross-reactive IgA, IgG, IgG1, IgG2a, IgG2b, IgG3, and IgM antibodies [41]. OMVs of multiple serotypes (MOMVs) including *S. dysenteriae* 1 Δstx, *S. flexneri* 2a, 3a and 6, *S. boydii* type 4 and *S. sonnei* triggered Th1/Th2 cell-mediated immunity in BALB/c mice and provided cross-reactive protection following infection with each individual serotype [42].

Single-serotype outer membrane vesicles (SMOVs) and MOMVs of S*. dysenteriae* 1, *S. flexneri* 2a, 3a and 6, *S. boydii* type 4, and *S. sonnei* induced the release of TNF-α, IL-1β, IFN-γ, IL-6, IL-10, and IL-4 [50]. Indeed, MOMVs induced immune responses more efficiently compared to SOMVs [50]. OMVs of *S. flexneri* coupled to nanoparticles (NPs) provided also protection against shigellosis in BALB/c mice. The oral and the nasal application of these NP-OMVs enhanced the release of IL-12p40 and of IL-10, and in particular, the oral vaccination protected against a lethal dose of *S. flexneri* [43]. Furthermore, individual components of OMVs such as OmpA enhanced the production of IFN-γ and TNF-α. When coupled to histaq as an adjuvant, the intranasal administration of OmpA also protected 100% of the challenged mice against shigellosis [44]. Further analyses indicated that the combination of OmpA from *S. flexneri* 2a with Freund’s complete adjuvant also protected mice against a lethal dose of *S. flexneri* 2a [45]. Next to TNF-α and IFN-γ, OmpA also enhanced the production of IL-6 and MIP-2α in this study. Furthermore, the intraperitoneal or subcutaneous application of OmpC from the *S. flexneri* serotype 3a coupled with monophosphoryl-lipid [51] as adjuvant provided variable protection against a lethal dose of *S. flexneri* [46].

Genetic modifications of individual OMV components represent another important strategy to enhance OMV-specific immune responses. For example, the subcutaneous administration of generalized modules of membrane antigens (GMMAs) along with Freund’s adjuvant (FA) or with aluminum hydroxide (alum) elicited the production of specific IgG antibodies in outbred CD1 mice [47]. A subsequent study confirmed the immunogenic potential of GMMAs in mice [48]. OMVs obtained from *S. boydii* strains with a tol-A mutation, one of the genes of the *Shigella* membrane, also induced the production of mucosal IgG and IgA antibodies and of inflammatory cytokines including TNF-α, IL-6, and IFN-γ. Importantly, these OMVs also elicited 100% protection against shigellosis in neonatal Swiss mice [41]. The Pan-Shigella surface protein 1 (PSSP-1) of the *S. flexneri* serotypes 2a and 6 as well as of *S. dysenteriae* serotype 1 strain is known as a potent immune-modulating agent. It enhanced the production of several cytokines including IL-17A, IL-2, IL-6, IL-4, TNF-α, and IFN-γ, and protected against shigellosis in murine infection models. Furthermore, the application of PSSP-1 along with the cholera toxin (CT) and a double mutant (R192G/L211A) of the heat-labile *E.coli* toxin (dmLT) triggered IgG-specific antibody responses [49].

## 6. Immunoreactive Proteins in OMVs

The outer membrane proteins (OMPs) of OMVs derived from Gram-negative bacteria induce potent immune responses. Thus, they are important targets for the development of vaccines. Indeed, several proteins obtained from *Shigella* OMVs have been identified and characterized. We have summarized the findings on the immune-reactive properties of some of the outer membrane proteins (Table 2) as well as on surface proteins and polysaccharides.

A 34 kDa major outer membrane protein (MOMP) of the *S. flexneri* serotype 2a enhances bacterial attachment to macrophages. It also augments the production and the release of nitric oxide and of various cytokines and chemokines including granulocyte-colony stimulating factor (G-CSF), IL-12p70, TNF-α, IL-6, MIP-1α, MIP-1β, and RANTES [52,53]. Furthermore, the 34 kDa MOMP promotes the expression of the Toll-like receptor (TLR2) as well as of MHC class II and CD80 molecules on macrophages, and thus, increases their stimulatory capacity. MOMP also activates p38 MAP kinase signaling and promotes the translocation of NF-κB to the nucleus, affecting the transcription and expression of other genes [53]. Similarly, the 24 kDa and 57 kDa MOMPs of *S. flexneri* also enhance the production of nitric oxide, TNF-α, and IL-12 in mouse macrophages, and thus, promote Th1 responses [54,59].

OmpA plays a pivotal role in the pathogenesis of *Shigellae* as described above. Furthermore, OmpA is highly immunoreactive. Thus, OmpA increases the secretion of IgG and IgA, and induces Th1 cells in mice. OmpA also activates macrophages and B cells, and enhances the expression of MHC class II, CD80, and CD40 molecules on both cell subsets. Furthermore, it augments the production of Th1 cytokines. Importantly, OmpA protects mice against a lethal dose of *S. flexneri* [44]. Similar as described for the 34 kDA MOMP, TLR2 engagement mainly mediates these immunological effects of OmpA [45]. In contrast to the 34 kDA MOMP, OmpA engages protein tyrosine kinase, ERK, and NF-κB as signaling pathways [56].

Human umbilical cord sera react to OmpC, a 38-kDa outer membrane protein of the *S. flexneri* serotype 3a. This reactivity has been associated with protective activity [46].

Interestingly, the Omps of one *Shigella* species can elicit cross-reactive immune responses against other *Shigella* species, as shown, for example, for OmpA and OmpC of *S. dysentariae* serotype 1 [60]. However, Omps do not only provide cross-species protection, but can also elicit immune responses across genera. Thus, the recombinant LacVax^®^ OmpA of *Lactococcus lactis* (*L. lactis*) induced antibody- and immune cell-mediated protection against *Shigella* infections [61].

EpiMix^®^, the chemically synthesized conserved epitopes of OmpA and of OmpC of *S. flexneri*, exhibit epitope-specific antibody responses following intramuscular administration in mice. Specifically, EpiMix^®^ stimulated the secretion of specific IgA and IgG antibodies in the blood, and the release of IL-4, IL-2, and IFN-γ in the feces [58].

Pan-Shigella surface protein 1 (PSSP-1), a conserved *Shigella* protein, also provides immunological protection against various *Shigella* species including *S. flexneri* serotypes 2a, 5a, and 6 as well as against *S. boydii*, *S. sonnei*, and *S. dysenteriae* serotype 1. Specifically, PSSP-1 induces local and systemic antibody responses as well as the production of IL-17A and IFN-γ [49].

Furthermore, polysaccharides might be interesting targets for vaccination as they are often the main components of the outer cell wall in various Gram-negative bacteria. Indeed, the mucosal delivery of polysaccharide antigens via OMVs induced protection against *S. flexneri* infection in mice [43]. The LPS of *Shigella flexneri* is a hexa-acylated isoform that can trigger optimal inflammatory activity. However, *Shigellae* drastically change the degree of acylation of the lipid A component of LPS during invasion into the host cell. The purified hypo-acylated LPS exhibits a reduced inflammatory potential, which allows the bacteria to lower the sensory activity of the immune system and to escape the downstream anti-bacterial effector mechanisms [62]. Serum IgG antibodies against LPS antigens also correlate with effective protection against shigellosis [63]. This is the reason why LPS or parts of LPS are utilized as components in conjugate compounds [64].

However, the structure of LPS, especially the O antigen, is diverse between individual *Shigella* serovars. Thus, the cross-reactivity of existing antibodies is likely to be limited. However, LPS from locally endemic circulating strains can trigger cross-reactivity in sera of patients infected with non-local *Shigella* spp. (unpublished observations). Interestingly, the O-specific polysaccharides with their preserved heptosis or Kdo residuals presumably preserve cross-reactivity to other *Shigella* spp. [65].

## 7. Virulence Factors of *Shigella* OMVs

*Shigella* species, like other Gram-negative bacteria, secrete OMVs during growth in vitro and in vivo. OMVs help to establish bacterial pathogenesis. They also deliver virulence factors including toxins into host cells [30]. We have summarized several virulence factors and toxins that *Shigella* spp. release in Table 3.

In detail, MxiD, a 64 KDa outer membrane protein (membrane-associated ring-forming protein, MxiD), for example, perpetuates the secretion of the Ipa invasins (IpaA, IpaB, and IpaC) of *S. flexneri*, which are pivotal for bacterial invasion into epithelial cells of the host [66]. MxiD is also an important component of the needle complex of the type III secretory machine of *Shigellae* [74,85].

Furthermore, the outer membrane lipoprotein, MxiM, plays a pivotal role in the invasion of *Shigellae*. Similar to MxiD, MxiM also acts though the type III secretion system [74,76]. MxiM also stabilizes MxiD in its location and both MxiD and MxiM contribute to the assembly of the type III secretion system machinery and the needle complex [74]. Another lipoprotein, MxiJ, plays a key role in the invasion process due to the secretion of the Ipa invasins, IpaA, IpaB, and IpaC of *Shigellae* [75].

A 35 KDa outer membrane protein A [55] is involved in IcsA exposition, cell-to-cell-spread, and protrusion formation [67]. OmpA plays also a vital role in other aspects of bacterial pathogenesis such as biofilm formation, host cell invasion, intracellular spread, the evasion of immune defences, and the mediation of cytokine production. In addition, it modifies host cell survival and mitochondrial fragmentation. Furthermore, OmpA serves as a receptor for bacteriophages [86,87,88,89].

The outer membrane protein IcsA (VirG) is an adhesin-like autotransporter protein of *Shigella* that is regulated by the type III secretion system. It is required for the pathogenesis of *Shigella* [90]. IcsA promotes the polymerization of actin. Thus, it is involved in the actin-mediated intra- and intercellular transmission of *Shigella* spp. in the host [69,70,71]. Furthermore, IcsA mediates biofilm formation and bacterial cell–cell communication [72].

SopA, the outer membrane protease of *Shigella*, is required for the polar localization of IcsA and the actin-based motility inside infected cells [68]. SopA, along with other effector proteins of the pathogenicity islands of *Shigella* is also responsible for the invasion of epithelial cells [91] and the induction of diarrhea [92].

YaeT (Omp85) is required for the secretion and the expression of the autotransporters IcsA and SepA of *S. flexneri* [77]. It acts as a Stx phage recognition site, mediates the dissemination of Shiga toxins, and the insertion of proteins in the outer membranes of Enterobacteriaceae [93,94,95].

The outer membrane protease (IcsP) cleaves the IcsA protein, and thus, controls the quantity and the distribution of IcsA. It also has an important role in the actin-based spread of *Shigella* in host tissues [68,80,81,82,83,84].

Cardiolipins are acidic diphosphatidylglycerols encoded by the synthase ClsA. They are involved in the surface localization of IcsA and the spread of *Shigella* [78]. Cardiolipins also promote the transportation of LPS to the outer membrane in Gram-negative bacteria [96].

Furthermore, the outer membrane phospholipase A (OMPLA), PldA and the β-barrel proteins in the outer membrane are responsible for the structural stability of *Shigella* membranes and type III secretion [79,97].

Thus, in summary, multiple virulence factors of *Shigella* spp. accumulate in OMVs. Thus, OMVs are presumably even more immunogenic compared to whole bacteria themselves.

## 8. Methods for Enhancing *Shigella* OMV Release and Yield

Bacteria produce OMVs during growth or infection in vitro and in vivo. The main hurdle in the isolation of OMVs is their low yield, as bacteria release only small quantities of OMVs. To tackle this problem, different approaches to increase the production and release of OMVs have been described. These include genetic manipulation, nutrient depletion [36,37,48,98,99,100,101,102], oxidative stress [103], heat shock exposure [104,105] or induction by detergents [106,107], antibiotics [108,109,110], peptides [31,111] or sonication [112,113,114]. Although these approaches are effective and enhance the induction and release of OMVs, they influence the size and the chemical composition of OMVs, and subsequently, the cellular interaction and immune response [31,105,107,110,112,115,116,117,118,119,120].

With respect to genetic manipulations, the release of OMVs in *S. boydii* serotype 4 strains was enhanced by about 60% due to a disruption of the *tolA* gene in the outer membrane. The released OMVs stimulated the secretion of mucosal IgG and IgA and of TNF-α, IL-6, and IFN-γ in immunized mice [38]. Disruption of the Tol-Pal system in *S. flexneri* ΔtolR mutants even resulted in an 8-times increase in the release of OMVs. The produced OMVs still induced specific antibodies and the expression of MHC-II or CD40 [121].

The evolution of generalized modules for membrane antigens (GMMA) due to the deletion of the *tolR* gene has been utilized to increase the release of OMVs in *S. sonnei*. GMMA are strongly immunogenic, but less toxic; thus, they appear to be suitable for use in human trials [48]. Another study also showed that mutations in *tolR* and *galU* enhanced the production of GMMA [47]. Mutations in the *virK* gene of *S. flexneri* also increased the production of OMVs in a temperature-dependent manner [122].

## 9. Augmenting OMV Immune-Reactivity

To utilize OMVs as targets for vaccination, their immunological properties can be enhanced due to the application of different methods (Table 4).

One way is to mix OMVs from various *Shigella* species, which results in multi-serotype outer membrane vesicles (MOMVs). MOMVs have been generated from a mixture of OMVs from *S. dysenteriae* 1 Δstx, *S. flexneri* 2a, 3a and 6, *S. boydii* type 4 and *S. sonnei*. In mice, an immunization with MOMVs increased antibody responses, and subsequently, protection against several *Shigella* serotypes. In addition, mice exhibited a consistent broad-spectrum antibody response, and thus showed protection against all tested serotypes [42]. Similarly, MOMVs produced from several serotypes significantly enhanced cytokine production in rabbits following immunization [50].

Chemical tools can also be utilized to increase the immunological activation potential of OMVs. A treatment of *S. flexneri* 2a with binary ethylenimine [123] led to the production of OMVs with increased immunological properties. Due to the inactivation of *S. flexneri*, it can be used as safe OMV-based vaccine [39,124,125].

Coating OMVs with various agents also enhances their immunological potential, stability, and safety. OMVs from *S. flexneri* were formulated in nano-capsules and applied to mice, which resulted in high and long-lasting immune protection [39]. In another study, Chitosan-TPP coated with an enteric polymer (NP-OMVs) from *S. sonnei* were highly efficient in triggering persistent immunological responses in mice [126].

## 10. Conclusions

OMVs exhibit a broad range of functions that are fundamental for the biology and the ecology of Gram-negative bacteria. These functions include, for example, bacterial cell–cell communication, growth, survival, and pathogenesis as well as the transmission of virulence factors, horizontal gene transfer or the delivery of toxins. Furthermore, due to the assembly of various antigenic structures, OMVs are also key factors for the induction of inflammatory responses in the host, and thus, they are attractive targets for vaccination. Furthermore, due to mixing OMVs from different *Shigella* serotypes, cross-reactive immunity can be induced. Therefore, the characterization of the biological role of OMVs and an understanding of their bio-generation as well as the identification of local circulating *Shigella* strains and serotypes is essential to establish them as essential tools to combat bacterial pathogens.

## Figures and Tables

**Figure 1 ijms-23-00994-f001:**
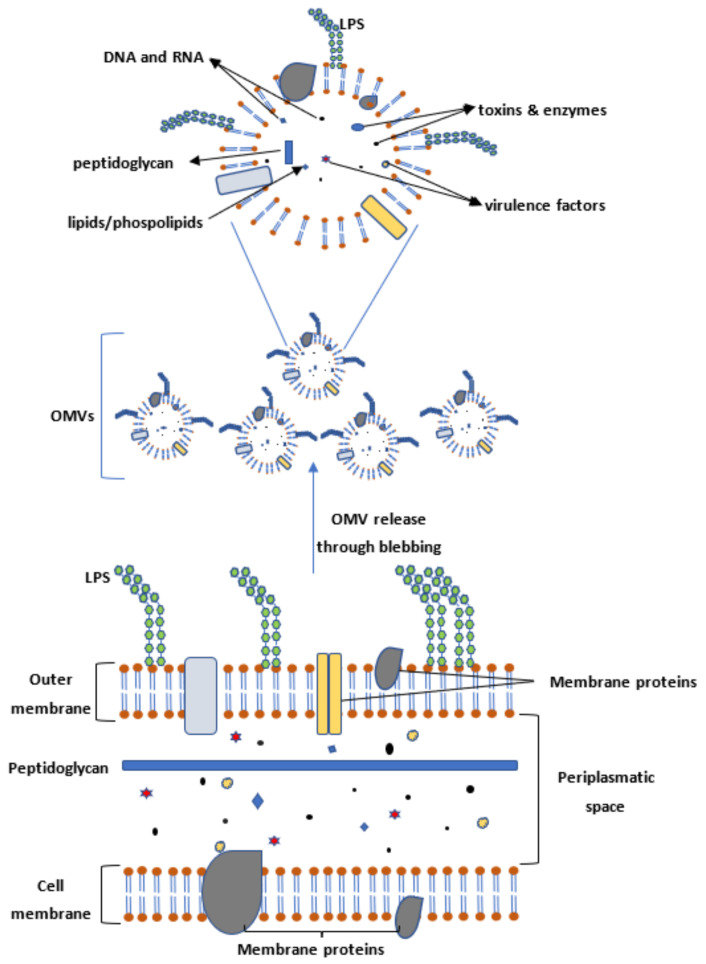
OMVs originate from the outer membranes of Gram-negative bacteria. They are 50–300 nm large vesicles that contain lipids, proteins, phospholipids, peptidoglycan, toxins, enzymes, LPS, DNA, and RNA derived from the periplasm between the inner membrane and the outer membrane as well as from the inner and outer membranes themselves.

**Figure 2 ijms-23-00994-f002:**
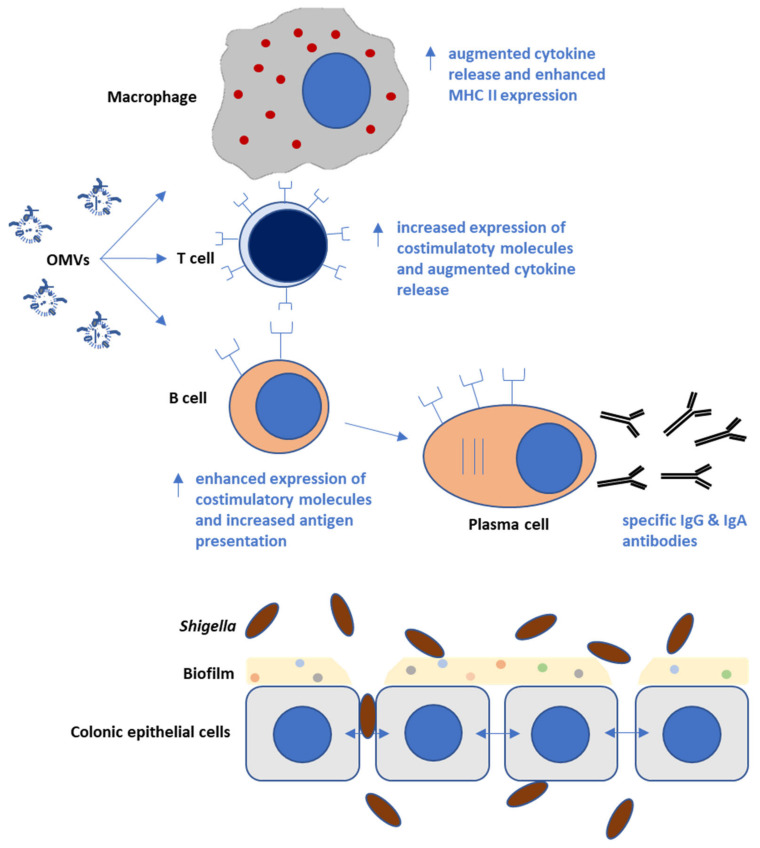
*Shigella* OMVs induce the release of different cytokines, chemokines, and specific antibodies by different immune cells, including macrophages, B and T lymphocytes. Their immune-stimulatory properties make OMVs attractive targets for vaccination against shigellosis.

**Table 1 ijms-23-00994-t001:** Evaluations of immunological response triggered by *Shigella* OMVs in infection models.

*Shigella* Species	Dose and Route of OMVs	Infection after Immunization	Duration of Protection	Protection	Ref
*S. flexneri* 2a	20 µg OMVs i.n.	35 days	0–15 days	100%	[39]
*S. flexneri* 2a	20 μg OMVs i.n./p.o.	28 days	0–9 days	50%	[40]
*S. dysenteriae 1*, *S. fle-neri* 2a, 3a and 6, *S. sonnei*.	32 μg OMVs p.o.	21 days	0–120 days	Variable	[41]
*S. dysenteriae* 1 Δ*stx*, *S. flexneri* 2a, 3a and 6, *S. boydii* type 4, *S. sonnei*	50 μg MOMVs p.o.			100%	[42]
*S. flexneri*	20–100 μg OMVs i.d. or p.o.	35 days		20–100%	[43]
*S. flexneri* 2a (N.Y-962/92)	3 μg recombinant his-tag OmpA i.n.	28 days	14 days	100%	[44]
*S. flexneri* 2a	1 μg OmpA i.p.	28 days	0–14 days	100%	[45]
*S. flexneri* 3a	1.6–20 μg OmpC s.c.	0–21 days		Variable	[46]
Genetically modified *S. sonnei*	0.2–2 µg GM-MA s.c.	0–35 days	0–49 days		[47]
*S. sonnei* 1790GAHB	29–238 µg GM-MA i.p.	21 days			[48]
*S. boydii*	25 μg tolA-disrupted OMVs p.o.	54 days		100%	[38]
*S. flexneri* 2a and 6, *S. dysenteriae 1*	20 μg PSSP1 i.n.	28 days	0–10 days	Variable	[49]

Note: ND, not determined; MOMVs: Multi-serotype outer membrane vesicles; SMOVs: Single-serotype outer membrane vesicles; GMMA: Generalized modules of membrane antigens; PSSP-1: pan-Shigella surface protein 1; NP: Nanoparticle; FA: Freund’s adjuvant; Alum: aluminum hydroxide; LD50: Infectious dose 50; LD: Lethal dose; MPL: Monophosphoryl lipid A; CT: cholera toxin; dmLT: double mutant (R192G/L211A) of heat-labile toxin of *E.coli*; i.n.: intranasal; i.p.: intraperitoneal; s.c.: subcutaneous; p.o.: per os/oral; i.d.: intradermal

**Table 2 ijms-23-00994-t002:** Immune-reactive proteins in OMVs from *Shigella*.

Immune-Reactive Protein	*Shigella* Species	Potential Role/Function	Ref
34 kDa major outer membrane protein (MOMP)	*S. flexneri* 2a	(i) Promotes binding to macrophages (ii) Increases the production of nitric oxide (iii) Enhances cytokine production	[52]
34 kDa MOMP	*S. flexneri* 2a	(i) Enhances TLR2 expression on macrophages. (ii) Increases the translocation of NF-κB to the nucleus (iii) Triggers the expression of p38 MAP kinases (iv) Augments the production of MyD88 and TRAF6 (v) Pormotes cytokine and chemokine production	[53]
OmpA	*S. flexneri* 2a	(i) Enhances the secretion of IgG and IgA (ii) Activates Th1 cells & macrophages (iii) Induces the expression of MHCII, CD80, CD40 (iv) Promotes the production of cytokines	[45]
34 kDa outer membrane protein	*S. flexneri* 2a	(i) Enhances production of nitric oxide, (ii) Increases TNF-α and interleukin-12 production	[54]
Outer membrane protein A [55]	*S. flexneri* 2a	(i) Enhances protective immunity (mucosal and systemic) by protein specific IgG and IgA responses. (ii) Increases the production of IgA secreting cells	[44]
Pan-Shigella surface protein 1 (PSSP-1)	*S. flexneri* 2a and 6; *S. dysenteriae* 1	(i) Enhances local and systemic antibody responses (ii) Increases the production of interleukin 17A and gamma interferon.	[49]
38-kDa OmpC	*S. flexneri* 3a	Increases B-cell specific antigenic epitopes (based on modelling)	[46]
Outer membrane protein A [55]	*S. flexneri* 2a	(i) Enhances the production of IgG and IgA (ii) Induces IL-6 and IL-10 production (iii) Increases MHC II and CD86 expression on B cells (iv) Promotes the differentiation of B cells into antibody secreting plasma cells	[56]
Outer membrane protein A [55]	*S. flexneri* 2a	(i) Activates NF-κB (ii) Enhances the production of cytokines and of NO (iii) Stimulates the T cells to release IFN-γ and IL-2	[57]
EpiMix^®^	*S. flexneri*	(i) Increases the secretion of specific serum IgG (ii) Enhances IgA, IL-4, IL-2and IFN-γ levels in feces	[58]

**Table 3 ijms-23-00994-t003:** Virulence factors of *Shigella* OMVs.

*Shigella* Species	Virulence Factors	Putative Function(s)	Ref
*S. flexneri*	MxiD, an outer membrane protein (omp)	Secretion of the Ipa invasins (IpaA, IpaB, and IpaC,) of *S. flexneri*. MxiD is an essential component of the Ipa secretion apparatus.	[66]
	Outer membrane proteinA [55]	IcsA exposition, cell-to-cell-spread and protrusion formation	[67]
	SopA, outer mem-brane protease	Required for the polar localization of IcsA and the actin-based motility inside infected cells	[68]
	Outer membrane protein IcsA (VirG)	Promotes bacterial transmission from host cell to host cell, mediates actin filament nucleation and unidirectional actin-based motility of *Shigellae*	[69]
	Outer membrane protein IcsA (VirG)	Involved in the actin-based motility required for intra- and intercellular *Shigella* spread	[70]
	Outer membrane protein IcsA (VirG)	Intracellular and cell-to-cell spread through polymerization of actin. Phosphoryation of IcsA and subsequent modulation of LcsA function	[71]
	Outer membrane protein IcsA (VirG)	Responsible for biofilm formation and bacterial cell to cell contact	[72]
	Outer Membrane Lipoprotein, MxiM	Plays a role in *Shigella* invasion and in the type III secretion system	[73]
	Outer Membrane Lipoprotein, MxiM	Supports the stability and localization of MxiD, it is required for the assembly in cells	[74]
	MxiJ, a lipoprotein	Mediates the secretion of *Shigella* Ipa invasins (IpaA, IpaB, and IpaC)	[75]
	Outer membrane protein C (ompC)	Involved in the spread of *Shigella* in epithelial cells	[76]
	YaeT (Omp85)	Required for the secretion and expression of *Shigella* auto-transporters IcsA and SepA.	[77]
	Cardiolipin (Gene encoded on synthase ClsA)	Involved in the surface localization of IcsA and spread of *Shigella*	[78]
	Outer membrane phospholipase A (OMPLA)-PldA	Essential for membrane stability and integrity, and type III secretion	[79]
*S. dys-enteriae,* *S. flexneri*	Outer membrane protease IcsP	Modulates the quantity and distribution of IcsA; role in actin-based motility-based *Shigella* spread	[68,80,81,82,83,84]

**Table 4 ijms-23-00994-t004:** Methods for enhancing the release, the immunological efficiency and the safety of *Shigella* OMVs.

Method	Mechanism	Increase in OMV Release	Immunological Efficiency	Ref
Enhancing OMV release	Disruption of *tolA*, one of the genes of the Tol–Pal system of membrane	60%	Mucosal IgG and IgA, pro-inflammatory cytokines (TNF-α, IL-6, IFN-γ)	[38]
	Distruption in Tol-Pal system in outer membrane	More than 8-times	Enhanced production of anti-bodies and expression of MHC II and costimulatory molecules	[121]
	Development of GMMA by deletion of *tolR*	Economic and high yield	Highly immunogenic	[48]
	Null mutants of *tolR* and *galU*	High yield, increased production of GMMA	Highly immunogenic	[47]
	*virK* mutant enhance the	High yield, OMV over- production	ND	[122]
Enhancing efficiency	Mixing of OMVs from multiple Shigella species to obtain MOMVs	ND	Consistent broad spectrum antibody response and protection against all tested serotypes	[42]
	Mixing of OMVs from multiple *Shigella* species → MOMVs	ND	Significantly enhanced cytokine production compared to SOMVs	[38]
	Binary ethylenimine [123] treatment	ND	Good immunogenic properties of OMVs	[124,125]
	Nanoencap-sulation of the OMVs	ND	Long-term protection	[39]
	Heat-induced (HT) outer-membrane vesicles development	ND	Higher contents of some antigenic structures than classical OMVs	[40]

Note: ND. not determined.

## Data Availability

Not applicable.

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
