# Peer review of "Shigella Outer Membrane Vesicles as Promising Targets for Vaccination"

_ijms, 2022, doi:10.3390/ijms23020994_

Round 1
Reviewer 1 Report
Thank you for the opportunity to review this comprehensive review article. Authors have reviewed Shigella outer membrane vesicles (OMVs) as a target for needed vaccination possibility.
Article is well written, with good flow of information and understandable reading. I have just a few minor comments:
In chapter 3, more detailed information should be put regarding previous studies that investigated vaccination approaches, to further confirm the main idea of this review
Line 195 – you can put reference numbers instead Pastor et al and Camacho et al
All tables could be adjusted and aligned better, as well as be wider and more reader-friendly
Lines 313-341 – This part could be re-written to be more comprehensive, as it has a number of small unneeded paragraphs
Author Response
Please find the point by point reply attached

Reviewer 2 Report
- “the mortality rate of shigellosis can be high”
In what circumstances?
- Use oxford comma in the whole text
- “to commonly used frontline antibiotics”
Please specify
- Which is the species most prevalent in developed countries?
- “Importantly, these OMVs also elicited 100% protection against shigellosis”
Specify in mice model; specify against what species
- Why vaccines efficacious in mice are not used (after trails) in humans?
Author Response

(The authors gave the same response as above.)
